# Surgery for Sporadic Primary Hyperparathyroidism: Evolution over the Last Twenty Years in a Monocentric Setting

**DOI:** 10.3390/cancers15092581

**Published:** 2023-04-30

**Authors:** Francesco Giudici, Laura Fortuna, Edda Russo, Benedetta Badii, Francesco Coratti, Fabio Staderini, Alessio Morandi, Clotilde Sparano, Luisa Petrone, Fabio Cianchi, Giuliano Perigli

**Affiliations:** 1Department of Experimental and Clinical Medicine, University of Florence, Largo Brambilla, 6, 50135 Florence, Italyalessio.morandi@stud.unifi.it (A.M.);; 2Department of Biomedical, Experimental and Clinical Sciences Mario Serio, University of Florence, Largo Brambilla, 6, 50135 Florence, Italy

**Keywords:** parathyroid, primary hyperparathyroidism, parathyroid cancer, parathyroidectomy, surgery

## Abstract

**Simple Summary:**

In this study, we describe a single-center experience based on a prospectively recorded and updated database that embraces the entire evolution of parathyroid surgery in sporadic primary hyperparathyroidism. In detail, surgically treated patients with sporadic primary hyperparathyroidism were divided into two groups, based on intraoperative parathyroid hormone (ioPTH) application. We focused on evaluating the long-term outcome in terms of treatment, persistence, relapse and complications, the role and current indications of ioPTH, and the feasibility, reproducibility and safety of transoral parathyroidectomy. The analysis shows that the use of ioPTH with the rapid method could be ineffective in helping surgeons in primary operations, especially when ultrasound and scintiscan are concordant. The advantages obtained by not using intraoperative PTH are not only economic. Indeed, our data shows shorter operating and general anesthesia times and hospital stays, having an important impact on patient biological commitment.

**Abstract:**

The sporadic parathyroid pathology of surgical interest is primarily limited to lesions that are the cause of hormonal hyperfunction (primary hyperparathyroidism). In recent years, parathyroid surgery has evolved significantly, and numerous minimally invasive parathyroidectomy techniques have been developed. In this study, we describe a single-center and well-documented case series of sporadic primary hyperparathyroidism, surgically treated by a single operator in the Endocrine Surgery Unit of the Surgical Clinic of the University of Florence-Careggi University Hospital, recorded and updated in a dedicated database that embraces the entire evolutionary timeframe of parathyroid surgery. From January 2000 to May 2020, 504 patients with a clinical and instrumental diagnosis of hyperparathyroidism were included in the study. The patients were divided into two groups, based on the application of intraoperative parathyroid hormone (ioPTH). The analysis shows that the use of ioPTH with the rapid method could be ineffective in helping surgeons in primary operations, especially when ultrasound and scintiscan are concordant. The advantages obtained by not using intraoperative PTH are not only economic. In fact, our data shows shorter operating and general anesthesia times and hospital stays, having an important impact on patient biological commitment. Furthermore, the significant reduction in operating time makes it possible to almost triple the volume of activity in the same unit of time available, with an undeniable advantage for the reduction of waiting lists. In recent years, minimally invasive approaches have allowed surgeons to reach the best compromise between invasiveness and aesthetic results.

## 1. Introduction

The sporadic parathyroid pathology of surgical interest is essentially limited to the hyperfunctioning adenoma and hyperplasia (primary hyperparathyroidism). In fact, parathyroid cancer is one of the rarest causes of primary hyperparathyroidism (1%) and generally occurs with more severe hypercalcemia. First described by Sainton and Millot in 1933 [1], it is the least commonly observed endocrine neoplasm worldwide. In sporadic primary hyperparathyroidism, the correct preoperative diagnosis includes clinical evaluation, serum calcium, phosphorus and parathyroid hormone (PTH) dosage, ultrasound, and scintigraphic and radiological imaging. The Endocrine Surgery Units favor the concentration of patients in high volume centers, where multi-specialist collaboration and high surgical competence are present. This is relevant because the preoperative imaging is sometimes discordant and not sufficient to protect from the occurrence of multiglandular disease, such as in the case of multiple adenomas or hyperplasia [2]. The quick intraoperative parathyroid hormone (ioPTH) was proposed, and subsequently validated, as the ideal means to verify the efficacy of the procedure, given that its two to three minutes half-life allows a decay from the preoperative values to be evaluated in a very short time.

Parathyroid surgery has changed a lot in recent decades. At the end of the 1980s, the refinement of localization imaging (ultrasound and scintigraphy with sestaMIBI) made it widely possible to preoperatively discriminate hyperparathyroidism, due to uniglandular disease (over 85% of cases), from the rarer multiglandular disease [3]. Nowadays, in cases of preoperative diagnosis of uniglandular disease, a “focused” approach on the single pathological parathyroid is preferred, with a bilateral exploration of the four glands restricted to a minority of cases, reducing morbidity (risk of damage only on the ipsilateral recurrent nerve and parathyroids), limiting tissue dissection with less postoperative pain, and the consequent possibility of reducing the in-hospital length of stay.

Regarding surgical technique, apart from the traditional approach, numerous minimally invasive parathyroidectomy techniques have been developed and described over the past two decades. This category includes both the so-called “open” approaches (minimally invasive open parathyroidectomy-OMIP; mini incisional parathyroidectomy- MIP; radio-guided minimally invasive parathyroidectomy-RGMIP), and video-assisted approaches, such as video-assisted parathyroidectomy (minimally invasive video-assisted parathyroidectomy-MIVAP), parathyroidectomy video-assisted with a lateral approach (video-assisted parathyroidectomy for lateral approach -VAP-LA), and some purely endoscopic techniques (endoscopic parathyroidectomy -EP, by extracervical access, namely trans-oral or/and trans-axillar way) [4,5].

We describe a single-center experience based on a prospectively recorded and updated database that embraces the entire evolution of parathyroid surgery in sporadic primary hyperparathyroidism. In particular, we focused on evaluating the long-term outcome in terms of treatment, persistence, relapse and complications, the role and current indications of ioPTH, and the feasibility, reproducibility and safety of transoral parathyroidectomy [6].

## 2. Materials and Methods

The study is based on a monocentric case series of sporadic primary hyperparathyroidism surgically treated by a single surgeon at the Endocrine Surgery Unit of the Surgical Department of the University Hospital of Florence from January 2000 to May 2020. A total of 504 patients were prospectively included in the study. The clinical diagnosis of hyperparathyroidism was always confirmed in all patients by biohumoral assays, ultrasound, and scintigraphy.

All cases were collected and encoded in a database at the time of the treatment, constantly updated, and containing demographic, anamnestic, clinical, diagnostic, treatment, and follow-up data. Incidental (high levels of calcemic and/or PTH incidentally detected, and those with a parathyroid lesion found at thyroidectomy) or symptomatic presentation were evaluated. About symptoms and signs, the most frequently found were urinary lithiasis, and/or painful or osteoporotic bone fracture symptoms. Despite the potential involvement of other organs and systems, we mainly considered the two previously described main clinical pictures (skeletal and urinary). Patients affected by Multiple Endocrine Neoplasia (MEN) syndromes were not included.

### 2.1. Preoperative Diagnostic Tests

The typical hemato-urinary profile and the findings of cervical ultrasound and sestamibi-scan were recorded.

### 2.2. Treatment

The surgical procedures were divided into:-conventional: bilateral cervical exploration, visualization of all the parathyroid glands, and removal of the pathological one or ones.-open minimally invasive (MIP) or video-assisted minimally invasive (MIVAP): identification and focused removal of glands indicated as pathological. The video-assisted procedure has been gradually abandoned in the last ten years, not offering substantial advantages over the minimally invasive open one.-transoral approach was recently adopted, consisting of a totally endoscopic procedure through three accesses in the inferior oral vestibulum (TOEPVA: trans oral endoscopic parathyroidectomy vestibular approach) [7], by us modified by moving the central access into the submental dimple (H-TOEPSA: hybrid trans oral endoscopic parathyroidectomy submental approach) to ensure optimal cosmetic results.

The surgical specimen was fully pathologically investigated. The operating time was considered from the skin incision to skin suture, even in cases with ioPTH assay. It consists of serial samples of intact PTH at pre-established times: basal before incision, during the manipulation, at removal, at 10 m’ and 20 m’ and exceptionally, 40 m’ from the excision. Calcium and PTH were dosed at 6, 18 and 22 h after surgery. 

The most common postoperative complications were recorded (transient or permanent hypocalcemia; compressive or non-compressive hematoma; mono or bilateral paralysis of the vocal cords).

### 2.3. Follow-Up

After one week, all patients underwent clinical evaluation and serum calcium and PTH assay, repeated after one month. Consistently with the purpose of the study, we divided patients into two groups: -group A: patients operated adopting ioPTH assay;-group B: patients operated without ioPTH assay.

We performed a cost analysis obtained by adding the cost of 34 euros for the measurement of PTH on each individual intraoperative sample, to the cost of the occupation of the operating room (on average estimated at 15 euros per minute), and by deducting the costs of re-operations that the intraoperative PTH assay would potentially have avoided. All these costs were referred to the specific regional Disease Related Groups (DRG) for parathyroidectomy (a global refund for the in-hospital costs = 1410 euros), updated to 2022, which considers a hospital stay of fewer than 24 h with just one overnight stay.

### 2.4. Statistics

Student’s *t*-test was used for inter-group comparisons. The level of significance of 0.05 was used in all statistical comparisons.

## 3. Results

The baseline characteristics of the patients were listed in Table 1.

Group A

Family history of endocrinopathy was reported in eight patients (12.3%). In 38.4% of patients, the diagnosis was made incidentally, while in 56.9% the diagnosis resulted from clinical typical symptoms of hyperparathyroidism.

In three patients (4.6%), (two previously operated on elsewhere), there was persistent hyperparathyroidism.

Regarding diagnostic preoperative localization, the findings are shown in Figure 1 and Figure 2. No significant differences have been observed comparing US and Scintigraphy results between group A and B.

Cervical US was performed in 63 patients (97%). In 16% it did not visualize any gland, in 71% only one enlarged gland, in 10% two glands. and in 3% three enlarged parathyroid glands. In 41 (65%) the ultrasound revealed a concomitant thyroid disease.

SestaMIBI scan, performed in 62 patients (95.3%), demonstrated localized uptake in 72.6%, no uptake in 25.8%, and uncertain uptake in one case (1.6%).

Out of the 10 cases with negative ultrasound, five had positive SestaMIBI scan (50%) and out of 53 cases with positive ultrasound, the scan was concordant in 42 cases (79.2%).

A total of 21 patients (32.8%) underwent a minimally invasive procedure (MIP and MIVAP), while 44 (67.2%) underwent traditional parathyroidectomy.

The pathological examination confirmed 51 (78.5%) single adenomas, six patients with (9.2%) double adenomas, four patients (6.2%) with diffuse hyperplasia and four (6.2%) without pathological parathyroid tissue (but normalized postoperative PTH and calcemic response).

Post-operative complications occurred in seven (10.8%) patients: three temporary hypocalcemia, one hypocalcemic crisis regressed on the first day after medical treatment, one left vocal cord functional paralysis in pre-existing right paralysis (temporary tracheostomy, removed after one week), one bilateral late vocal cord paresis with dyspnea (tracheostomy on the eighth day, removed in the following month), 1 persistent dysphonia normalized after two months.

Persistence of disease was demonstrated in one patient (1.5%) who did not show ioPTH decay at 20′ and 40′ because of a missed second adenoma (PTH of 44.4 pmol/L on the 10th day). Another patient (1.5%) experienced persistence after a negative cervical exploration due to posterior adenoma. On the sixth day, the patient had a PTH value of 36.6 pmol/L, comparable to the preoperative value. The sestaMIBI scan performed two months later showed a deep posterior left uptake, confirmed by ultrasound.

Globally, the mean operative time was 112 min.

Considering the current average cost of using the operating room to be 15 euros per minute, the surgical procedure alone amounts to 1680 euros per patient to which, if the intraoperative PTH assessment is used, another 170 euros must be added for at least five PTH determinations with a unit value of 34 euros (*p* < 0.05).

Group B

For research purposes in 85 out of 439 patients of this group, serial PTH blood samples (at the same intra and post-operative times than in group A) were performed, but without waiting for their results (ready the day after surgery) to guide the surgeon.

There was a family history of endocrinopathies in 27 patients (6.1%).

In 261 patients (59.4%) the diagnosis was made incidentally.

In 178 patients (40.6%) the diagnosis resulted from clinical symptoms.

Regarding diagnostic preoperative localization, the findings are shown in Figure 1 and Figure 2.

Cervical US performed in 422 cases (96.2%) did not visualize any gland in 21.3%, visualized only one gland in 72.3%, two glands in 5.5%, and three glands in 0.9%. SestaMIBI scans, performed in 371 cases (84.5%) (Figure 2), demonstrated localized uptake in 298 (80.3%), no uptake in 63 (16.9%) and doubtful uptake in ten (2.7%).

Out of 90 patients with negative ultrasound, 42 patients (47%) had a diagnostic SestaMIBI scan and 48 (53%) a negative or doubtful scan. The other 332 patients with positive US had preoperative concordant sestaMIBI scans in 304 cases (91.5%).

Clinical data comparison between groups A and B was reported in Table 2.

Parathyroidectomy was performed in 151 cases (34.4%) with a mini-invasive MIP open or H-TOEPSA procedure, while 288 cases (65.6%) were treated with conventional parathyroidectomy.

Pathological examination showed 390 (88.8%) single adenomas, 11 (2.5%) double adenomas, 19 (4.3%) diffuse hyperplasia, six (1.3%) non-pathological tissue (but normalized postoperative PTH and calcemic response), seven (1.5%) atypical adenomas (one intrathyroidal) and six (1.3%) malignant parathyroid tumors.

Postoperative complications were found in 41 patients (9.3%): 18 (4.1%) temporary symptomatic hypocalcemia, resolved with Ca and Vit D3; 17 (3.8%) temporary dysphonia with normalization within 1 month, and six (1.3%) superficial hematomas treated with superficial tubular drainage.

Persistence of hyperparathyroidism was observed in six patients (1.3%): Five patients (0.9%) had double adenomas, missed at the first surgery, while one patient (0.3%) experienced failure to find the adenoma.

The mean operative time was 38 min.

Considering the current average cost of using the operating room (15 euros per minute) the surgical procedure amounts to 570 euros per patient.

## 4. Discussion

During the National Institutes of Health (NIH) consensus conference of 1991, the radiologist John Doppman stated that the only useful study for preoperative localization of pathological parathyroid glands was the identification of an experienced surgeon [8,9,10]. Thereafter, the 2002 NIH Consensus Development Conference Panel reached similar conclusions about the poor reliability of contemporary radiological imaging [11]. 

Since then, the diagnostic and therapeutic approaches of sporadic primary hyperthyroidism have witnessed critical advancements.

For instance, the constant improvement of scintigraphic techniques, coupled with the ever-increasing resolution power of US probes, allowed successful tailored treatments (i.e., up to 98% of cases), avoiding useless cervical explorations. Moreover, modern minimally invasive surgical procedures have been increasingly validated, becoming the actual standards of experienced endocrine surgeons [12,13,14,15]. In fact, the outdated obsolete intraoperative gland localization techniques have been replaced by the current minimally invasive surgical approaches, which require specialized and skillful tertiary centers.

The past intraoperative frozen sections showed poor accuracy due to the difficulty of distinguishing healthy tissue from hyperplastic or adenomatous tissue, despite still requiring gland excision.

On the contrary, the modern and almost completely endoscopic procedures, which are confined in a reduced working chamber, overcome the former limitations and the extra cervical access provided by our modified transoral access clearly exemplifies these recent advancements [16,17,18]. In fact, by these approaches the ocular vision and direct manipulation of structures are replaced by an indirect screen projection. These techniques could take advantage of the recent use of fluorescence of the vital green dye of indocyanine injected intraoperatively and which has provided promising evidence of efficacy in facilitating the identification of parathyroid glands [19,20,21].

For these reasons, rapid intraoperative dosing of PTH has long been identified as the ideal means to confirm the removal of the hyperfunctioning tissue. However, after a few years of widespread use, the confidence in its initial absolute accuracy was eroded by an increasing number of reports on false positives and false negatives approaching or even exceeding those of selected series of patients with the two preoperative imaging examinations in agreement, as demonstrated in 2004 by a Mayo Clinic series [22,23,24,25,26,27], but also from our previous experiences [28,29]. The continuous reduction of economic resources then made the availability of the equipment and of the dedicated technician in the operating room more and more difficult, furthermore, the sending of samples to the laboratory further extended the waiting times and consequently the costs, up to exceeding the limit of profitability. Finally, the large availability of refined imaging and the growing concentration of this pathology in dedicated centers, questioned the real usefulness of intraoperative PTH dosing, suggesting to narrow/limit its use to many selected cases.

The long period covered by the present study allowed us to observe the entire evolution of the diagnostics and surgical treatment of hyperparathyroidism [30]. A further strength of the present analysis is that all the patients were consecutively diagnosed and treated in the regional Endocrine Surgical Reference Center by almost exclusively the same team of specialists.

Despite the differences in sub-cohort sizes, due to the quite total drop out of intraoperative PTH in recent years, the two series seemed sufficiently representative and comparable for the considered variables. In fact, all the eligible patients underwent the same pre, intra and postoperative path and were checked with the same deadlines by the surgical team. This rigorous approach reflects the few cases of failure, which stands at only eight (1.5%) out of a total of 504 operated cases (about 98.5% cure).

In group A, intraoperative PTH was measured in 65 patients. The five false negatives cases (7.7%) required an unnecessary extension of surgical exploration to both thyroid lodges and to the sites of possible parathyroid ectopia. However, the possibility of a brief hospitalization within 24 postoperative hours, which included an overnight stay and kept patients out of the office, did not negate the risk of symptomatic hypocalcemia, which should be treated promptly with intravenous calcium administration.

Both the groups showed comparable levels of PTH and normalized calcium at the time of discharge, while the upper limit was found to be that of PTH but not of calcium at the control at thirty days. This phenomenon, which initially created some concern despite normal calcium, suggesting the unrecognized presence of a multi-glandular disease, is due to a delayed resetting of the cellular receptors for calcium in the residual healthy parathyroid glands. This impairment affected about a fifth of cases, but they spontaneously recovered during the medium follow-up.

Of the 385 patients with positive ultrasound, those 346 patients with concordant preoperative ultrasound (89.6%) were successfully treated except one with a double adenoma (Table 1).

Focusing on the main point of the study, it is evident the difference in the operating time between the two groups (112 m’ vs. 38 m’). The huge gap caused primarily by the use of ioPTH is also explained by the recent increased use of minimally invasive procedures that are much shorter than conventional ones, which were instead prevalent in the first part of the experience.

In the entire series, eight patients (1.5%) experienced surgical failure (Table 2). In 2 out of 65 (3.0%) group A patients, ioPTH had not decayed in the expected times. In one case, a 59-year-old male with a clinical and humoral preoperative diagnosis but negative imaging, the extension of the exploration was in any case ineffective in finding the lesion. It was instead detected in the left retro-jugulo-carotid region by repeated postoperative imaging. However, he was not reoperated for severe cardiological disease. In the other case, a 70-year-old female, preoperative imaging was concordant for the right lower adenoma, which was actually found, removed, and histologically confirmed; the extended exploration, induced by the unsatisfactory decay of PTH, had identified the other glands as normal in terms of location and morphology. The persistence of hyperparathyroidism led to the suspicion of a second adenoma, at the ectopic site, but the patient refused our recommended diagnostic re-evaluation. In neither case was the intraoperative dosage of PTH useful in increasing the chances of a cure.

In the group of 439 patients operated on without intraoperative PTH, 6 (1.3%) patients (five females and one male) did not achieve a cure at the initial attempt. In three of the five females, a first histologically-confirmed adenoma was initially removed and subsequently, a second adenoma was detected on postoperative re-imaging, which allowed the second successful surgery. In another female patient and in the male patient, persistence was attributed to the possible presence of a second adenoma after the histologically confirmed removal of the first. Both patients refused our reassessment and moved to another center. Finally, in the last female patient, who had weakly positive scintiscan on the right but negative pre- and intraoperative ultrasound, the severe scarring from a previous total thyroidectomy made cervical exploration extremely difficult, ending unsuccessfully, despite being systematically performed also at the potential ectopic sites. In the right loggia, where the scintiscan indicated an uptake, only one identifiable parathyroid was found, but with an absolutely normal appearance and location. The thyrothymic lymphatic tissue and the right thymic horn sent for histological examination did not reveal any parathyroidal tissue. The postoperative re-evaluation conducted by the endocrinologists hypothesized a reactive increase of PTH and questioned the correctness of the initial diagnosis of primary hyperparathyroidism. [31].

The use of the intraoperative PTH could have facilitated the achievement of the cure in 5 out of 6 patients, even if the non-decay of PTH is only predictive of residual hyperfunctioning tissue, not an instrument to localize it.

Evaluating the costs, in our setting, we can estimate 1850 euros/patient if the ioPTH is performed, and 570 if it is not, with a saving of 1280 euros/patient, which for the 439 cases in group B amounts to 561.920 euros, even after subtracting the costs of 8400 euros due to the reoperation of six (1.3%) initially uncured patients. Furthermore, we have to add the advantages resulting from the shorter general anesthesia times and hospital stays. The significant reduction in operating times (*p* < 0.05) would also almost triple the volume of activity with an undeniable advantage for the reduction of waiting lists.

Lastly, concerning the feasibility, reproducibility, and outcome of our modified transoral procedure, the few cases treated do not allow us to draw statistically significant conclusions. However, the initial results already allow us to foresee that, like it happened for thyroidectomy, the removal of the parathyroid glands via a hybrid transoral endoscopy would be proved to be the most suitable procedure to obtain the best compromise between invasiveness and cosmetic result.

## 5. Conclusions

The analysis of this series of patients operated on for sporadic primary hyperparathyroidism allows us to conclude that the use of the quick ioPTH should no longer be routinely applied. It should be reserved for limited patients in whom preoperative imaging is absolutely negative or can lead to suspicion of a rare multi-glandular disease, even if, in this case, the bilateral exploration would be mandatory anyway.

The usefulness during reoperation is less doubtful though because it can alleviate the difficulty for the surgeon in finding even normotopic parathyroid gland within the scarring fibrosis following the first surgery. Despite the evolution of imaging and surgical procedures, the operator’s skill and experience are still crucial, together in addressing primary treatment in high volume centers.

## Figures and Tables

**Figure 1 cancers-15-02581-f001:**
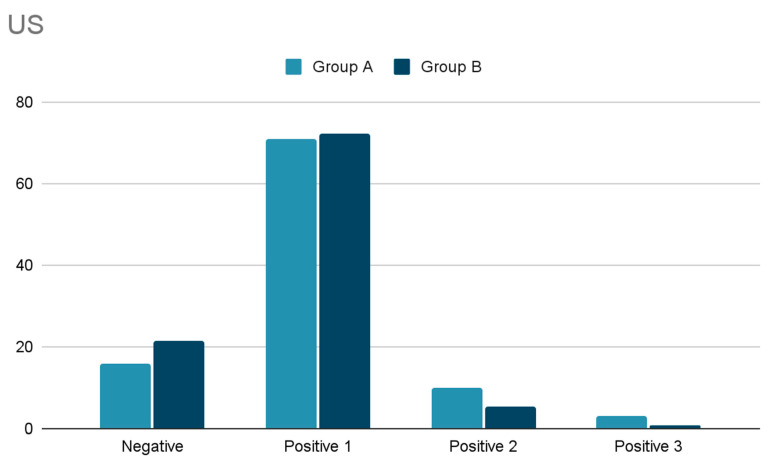
Comparison of US results between the group A and B (numbers are referred to the number of glands described).

**Figure 2 cancers-15-02581-f002:**
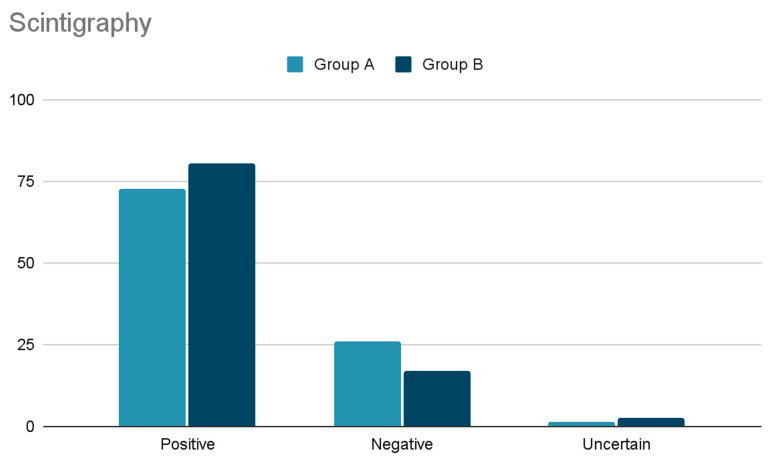
Comparison of Scintigraphy results between group A and B.

**Table 1 cancers-15-02581-t001:** Data comparison between groups A and B, preoperative.

	Group A (n = 65) IOPTH Group	Group B (n = 439)	*p* Value
** *Baseline* **			
*Gender (m/f%)*	20(30.9)/45 (69.2)	93 (21.2)/346 (78.8)	-
*Mean age (m/f)*	61.4/63.9	57.1/60	-
** *Onset* **			
*Asymptomatic (%)*	25 (38.4)	261 (59.4)	**0.046**
*Symptomatic (%)*	37 (56.9)	178 (40.6)	**0.033**
*Persistence (%)*	3 (4.6)		0.956
** *Preoperative imaging* **			
*US (%)*	63 (97%)	422 (96.2)	0.220
*Scintigraphy (%)*	62 (95.3)	371 (84.5)	**0.034**
*US −/Scint. + (%)*	10/5 (50)	90/42 (47)	0.096
*US +/ Scint. −(%)*	53/43 (79.2)	332/340 (91.5)	**0.031**
** *Preoperative blood tests* **			
*PTH (pmol/L)* (ref: 1.3–7.5)	32.95	32.8	0.092
*Calcium (mg/dL)* (ref: 8.6–10.2)	11.54	11.31	0.123
*Phosphorus (mg/dL)* (ref: 2.5–4.5)	2.2	2.4	0.501
*Calciuria (mg/24 h)* (ref: 100–300)	407.7	446.5	**0.046**
*Phosphaturia (mg/24 h)* (ref: 400–1300)	794.8	763.3	0.087

**Table 2 cancers-15-02581-t002:** Data comparison between groups A and B, operative.

	Group A (n = 65) IOPTH Group	Group B (n = 439)	*p* Value
* **Procedure** *			
*Mininvasive (%)*	21 (32.8) MIP-MIVAP	155 (34.4) MIP-H-TOEPSA	0.077
*Conventional (%)*	44 (67.2)	288 (65.6)	0.157
* **Mean operative time (min)** *	112	38	**0.022**
* **Histopathology** *			
*Unique adenoma (%)*	51 (78.5)	390 (88.8)	**0.043**
*Double adenoma (%)*	6 (9.2)	11 (2.5)	**0.030**
*Diffuse hyperplasia (%)*	4 (6.2)	19 (4.3)	0.167
*Nonspecific tissue (%)*	4 (6.2)	6 (1.3)	**0.041**
*Atypical adenoma (%)*	-	7 (1.5)	**0.023**
*Carcinoma (%)*	-	6 (1.3)	**0.045**
* **Postoperative complications** *	7 (10.8)	41 (9.3)	0.576
* **Outcomes** *			
*Persistence (%)*	2 (3)	6 (1.3)	0.096
*Remission (%)*	63 (97)	433 (98.7)	0.568
* **Follow-up** *			
*Mean 6 h Ca/PTH (mg/dL; pmol/L)*	9.9/1.75	10.5/2.4	0.267
*Mean 8–22 h Ca/PTH (mg/dL; pmol/L)*	8.9/4.3	8.9/3.5	0.654
*Mean 1 month Ca/PTH (mg/dL; pmol/L)*	8.9/9.5	9/9.7	0.454
* **Costs (euro)** *	1680 + 170	570	**0.01**

*p* < 0.05 = significant.

## Data Availability

The data that support the findings of this study are available under reasonable request to the corresponding author M.B.

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
