# Peer review of "Surgery for Sporadic Primary Hyperparathyroidism: Evolution over the Last Twenty Years in a Monocentric Setting"

_cancers, 2023, doi:10.3390/cancers15092581_

Round 1
Reviewer 1 Report
The two series in this study showed the entire evolution of the diagnostics and surgical treatment of hyperparathyroidism. Despite the considerable numerical diversity, these series are representative and comparable for the variables considered.
Author Response
We thank the reviewer for the positive comment.
Reviewer 2 Report
The authors used their single center database on sporadic hyperparathyroidism to compare several aspects of parathyroid surgery, mainly regarding with and without intra-operative PTH measurements. Main results is that, in general, intra-operative PTH measurements are not necessary.
After reading the manuscript several important issues remain:
1. General comment
A) Main problem with the current study is the main goal of the study is not well-described, and consequently this makes drawing conclusions difficult. Therefore, it is important to focus a bit more throughout the whole manuscript.
B) Please pay attention to the used English terms; e.g. ‘dosage’ is frequently used incorrectly.
2. Abstract
A) It seems that only the background and advantages of the study are described. It lacks results and a conclusion. Please rewrite.
3. Introduction
A) Lines 67-73; please illustrate the knowledge gaps better, and thereafter the purpose of the study. Just stating ‘we believe we can……answer’ is too vague.
4. Additional comments
A) Line 99-101; this is rather vague. Please rephrase.
B) Line 124-127; this should be in the Results section.
C) Table 1. It would advise the add the reference values of the blood tests. Further, ‘%’ are lacking in several rows.
D) Figure 1. This would be easier to understand when a caption is added explaining what ‘1’, ‘2’, and ‘3’ means.
Author Response
- General comment
- A) Main problem with the current study is the main goal of the study is not well-described, and consequently, this makes drawing conclusions difficult. Therefore, it is important to focus a bit more throughout the whole manuscript.
We thank the reviewer for the comment, we focused our goal throughout the whole manuscript
- B) Please pay attention to the used English terms; e.g. ‘dosage’ is frequently used incorrectly.
We corrected it in the manuscript
- Abstract
- It seems that only the background and advantages of the study are described. It lacks results and a conclusion. Please rewrite.
We thank the reviewer for the suggestion, we rewrote the abstract accordingly
- Introduction
- Lines 67-73; please illustrate the knowledge gaps better, and thereafter the purpose of the study. Just stating ‘we believe we can……answer’ is too vague.
We illustrated the knowledge gap better, in accordance with the suggestion of the reviewer
- Additional comments
- Line 99-101; this is rather vague. Please rephrase.
We rephrased the mentioned lines
- Line 124-127; this should be in the Results section.
We corrected in the text
- Table 1. It would advise the add the reference values of the blood tests. Further, ‘%’ are lacking in several rows.
We added reference values and ‘%’
- Figure 1. This would be easier to understand when a caption is added explaining what ‘1’, ‘2’, and ‘3’ means.
We corrected in the caption
Round 2
Reviewer 2 Report
The authors answered the questions and changed the manuscript accordingly.
Just one minor point; please pay attention to the used English language.
Author Response
We thank the reviewer for the comment, we reviewed again the manuscript.